# Latent Segmentation of Stock Trading Strategies Using Multi-Modal Imitation Learning

**Iwao Maeda** [1,*]**, David deGraw** [2]**, Michiharu Kitano** [3]**, Hiroyasu Matsushima** [1] **, Kiyoshi Izumi** [1]**,**
**Hiroki Sakaji** [1] **and Atsuo Kato** [3]

[1]   Department of Systems Innovation, School of Engineering, The University of Tokyo, Tokyo 113-8654, Japan;
      hiroyasu-matsushima@biwako.shiga-u.ac.jp (H.M.); izumi@sys.t.u-tokyo.ac.jp (K.I.);
      sakaji@sys.t.u-tokyo.ac.jp (H.S.)
[2]   Daiwa Securities Co., Ltd., Tokyo 100-6752, Japan; david.degraw@daiwa.co.jp
[3]   Daiwa Institute of Research Ltd., Tokyo 135-8460, Japan; michiharu.kitano@gmail.com (M.K.);
      atsuo.kato@dir.co.jp (A.K.)
*   Correspondence: d2018imaeda@socsim.org

**Abstract:** While exchanges and regulators are able to observe and analyze the individual behavior of financial market participants through access to labeled data, this information is not accessible by other market participants nor by the general public. A key question, then, is whether it is possible to model individual market participants' behaviors through observation of publicly available unlabeled market data alone. Several methods have been suggested in the literature using classification methods based on summary trading statistics, as well as using inverse reinforcement learning methods to infer the reward function underlying trader behavior. Our primary contribution is to propose an alternative neural network based multi-modal imitation learning model which performs latent segmentation of stock trading strategies. As a result that the segmentation in the latent space is optimized according to individual reward functions underlying the order submission behaviors across each segment, our results provide interpretable classifications and accurate predictions that outperform other methods in major classification indicators as verified on historical orderbook data from January 2018 to August 2019 obtained from the Tokyo Stock Exchange. By further analyzing the behavior of various trader segments, we confirmed that our proposed segments behaves in line with real-market investor sentiments.

**Keywords:** neural networks; latent segmentation; imitation learning

## 1. Introduction

Research into stock traders' behavior patterns and operating principles is important to understand the dynamics of financial markets Brock et al. (1992); Obizhaeva and Wang (2013). Many studies have investigated stock trading strategies using dynamic models Ladley (2012); Rust et al. (1992); Vytelingum et al. (2004), and neural network models Chavarnakul and Enke (2008); Chen et al. (2003); Krauss et al. (2017). However, these models cannot explain traders' behavior perfectly, and more sophisticated methods are desired.

However, the general problem of modeling market agent behavior from historical data is complicated by the sheer number of agents and the diversity of their utility functions. It is simply impractical to try to enumerate them all. Furthermore, most stock trading data are anonymized Comerton-Forde and Tang (2009), and information about who submitted a certain order is not available.

The goal of this study is to learn stock trader behavior patterns from anonymized historical stock order data using neural network-based imitation learning (IL) Schaal (1999); Schaal et al. (2003). To realize

multi-modal imitation learning, we propose latent segmentation Cohen and Ramaswamy (1998); Swait (1994) of stock trading strategies by trader objective function. Orders are segmented according to the weighted average of the reward function for each stock trader segment. An IL model is defined for each segment and trained to predict which trader segment was most likely to have submitted a particular order at a particular time. We refer to the proposed method as "Latent Segmentation Imitation Learning (LSIL)".

LSIL was evaluated using both simulated market data and actual historical stock order data. Experiments using simulated data were conducted to evaluate the validity of latent segmentation, and experiments on historical stock order data were conducted to examine the accuracy of stock order predictions made by LSIL. We find that LSIL models are able to predict stock orders with a degree of accuracy, and also provide meaningful insights into the drivers of trader behavior. Detailed investigation into changes in market conditions and segments revealed that our proposed segments behaves in line with real-market investor sentiments.

The primary contributions of this study can be summarized as follows:

1. We propose a neural network-based method for imitation learning of stock trading strategies. To consider diverse trading strategies, latent segmentation of based on a reward function is introduced.
2. The proposed method is evaluated using both simulated market data and historical stock order data. The proposed method was confirmed to provide both accurate stock order predictions and a meaningful interpretation of trader segment behavior.

## 2. Related Work

### 2.1. Modeling Stock Trading Strategies

Modeling stock trading strategies have a long research history. Traditionally, dynamic model-based approaches apply simple rules and formulas to describe trader behavior. For simulated markets, various agents that mimic certain idealized real-life trader behavior are defined, such as random traders Raberto et al. (2001) as well as value and momentum traders Muranaga and Shimizu (1999). There are also strategies derived from the financial engineering perspective such as the sniping strategy Rust et al. (1992), the Zero intelligence strategy Ladley (2012), and the risk-based bidding strategy Vytelingum et al. (2004). More recently, as well as the development of rule-based strategies Chen et al. (2019); Yu et al. (2019), neural network-based methods Xiao et al. (2017) have been applied to learn strategies and develop models based on observed data. For example, deep reinforcement learning (DRL) has been applied in financial markets Meng and Khushi (2019) as well as deep direct reinforcement learning (RL) Deng et al. (2016) and DRL for price trailing Zarkias et al. (2019) have been proposed.

### 2.2. Imitation Learning

Imitation learning (IL) has been studied extensively in applications to robotics, navigation, game-play, and other areas Hussein et al. (2017). Recently, neural network-based IL methods, such as one-shot IL Duan et al. (2017); Finn et al. (2017), third-person IL Stadie et al. (2017), and Generative adversarial IL (GAIL) Ho and Ermon (2016) have been proposed. Inverse RL has also seen active interest in applications involving learning expert behavior. Abbeel and Ng (2004); Ng and Russell (2000). Some imitation leaning methods have been applied to financial tasks Liu et al. (2020).

In related work, other multi-modal IL methods have been proposed Kuefler and Kochenderfer (2018); Piao et al. (2019). Hausman et al. (2017) proposes a method for training multi-modal policy distributions from unlabeled data using generative adversarial networks (GAN) Goodfellow et al. (2014). However, combinations of multiple policies derived from multiple participants have yet to be studied.

*2.3. Latent Segmentation*

Latent segmentation is a method to partition data based on statistical information Cohen and Ramaswamy (1998), and is primarily seen in marketing approaches based on consumer segments Bhatnagar and Ghose (2004); Swait (1994). In recent years, latent segmentation using a deep neural network has been proposed and applied to various tasks Angueira et al. (2019); Ezeobiejesi and Bhanu (2017); Nguyen et al. (2018); Villarejo Ramos et al. (2019).

## 3. Latent Segmentation Imitation Learning (LSIL)

As mentioned previously, financial markets consist of stock orders with various objectives, and these objectives are not self-evident from trading data alone. Since previous studies were seemingly able to classify trading strategies Yang et al. (2015), we should be able to achieve higher prediction accuracy by modeling each strategy class. In this study, we assume latent segmentation of traders Cohen and Ramaswamy (1998); Swait (1994) and that all traders belong to a unique segment at each time, that each trader may drift between segments, but cannot belong to more than one segment simultaneously.

Let the latent segments $s_i$ $(i = 1, 2, \ldots, N)$ represent specific trading strategies. Then, the probability of submitting stock order $\pi(o|X)$ can be written as

$$\pi(o|X) = \sum_{i=1}^{N} p(s_i|X)\pi_{s_i}(o|X; \theta_i) \tag{1}$$

where $X$ are market states, $p(s_i|X)$ is the probability that traders belonging to segment $s_i$ submit an order, and $\pi_{s_i}(o|X; \theta_i)$ is the probability that a stock order is submitted by traders in segment $s_i$ conditioned on parameter $\theta_i$. Each $\pi_{s_i}(o|X; \theta_i)$ is predicted using an individual network, which we refer to as segment level order networks.

Although we can obtain $(X_t, o_t)$ pairs from historical stock order data, information about $s_i$ is never available. Therefore, we also predict $p(s_i|X)$ using another neural network that we refer to as the segment network. The predicted probability of a given segment is written $p(s_i|X; \theta_s)$, where $\theta_s$ is the parameter of the segment network.

As mentioned previously, each segment represents an individual strategy. Much like in reinforcement learning, we introduce an individual reward function for each segment. The reward function for segment $s_i$ is denoted $r_{s_i}(o)$. Then, with the predicted segment probability, the expected reward for order $o$ is calculated as follows.

$$E[r(o)] = \sum_{i=1}^{N} p(s_i|X; \theta_s)r_{s_i}(o) \tag{2}$$

Since real markets are not perfectly efficient Jung and Tran (2016), we assume there exists a segment of traders that acts inefficiently. We introduce an exceptional action segment $s_*$, which has a uniform reward function and requires $\sum_{i=1}^{N} p(s_i|X; \theta_s) + p(s_*|X; \theta_s) = 1$. $s_*$ represents noise traders Shleifer and Summers (1990) who do not act efficiently, and/or traders whose investment strategies do not fit our segmentation scheme. The gradient with respect to the parameters in the segment network $\theta_s$ is calculated as follows:

$$-\nabla_{\theta_s} \left( \sum_{i=1}^{N} p(s_i|X; \theta_s)r_{s_i}(o) + p(s_*|X; \theta_s)r_* \right) \tag{3}$$

Appropriate selection of reward function $r_{s_i}(o)$ is essential for training the segment network. Reward functions across the segments are constrained to similar scales in order for training to converge. In this study, as the simplest case, $r(o)$ is calculated as the profit and loss (P&L) of the order $o$. Thus, $r(o)$ can be calculated as follows.

$$r(o) = p_{mid,\emptyset}\Delta I + \Delta c \tag{4}$$

Here, $p_{mid,\emptyset}$ is the mid price of the stock at time $\tau$ after order $o$ is submitted. $\Delta I$ and $\Delta c$ are change in inventory and cash due to order $o$ and can be calculated from the change of the market limit-order-book. For LMT and MKT, $\Delta I$ is the executed quantity and $\Delta c$ is the transaction amount. $\Delta I \geq 0$ and $\Delta c \leq 0$ in case of buy LMT or MKT, and $\Delta I \leq 0$ and $\Delta c \geq 0$ in case of sell LMT or MKT. For CXL, $\Delta I$ and $\Delta c$ are excluded quantity and transaction amount; $-1$ times executed quantity and transaction amount of the canceling order. Therefore, differing from LMT and MKT, $\Delta I \leq 0$ and $\Delta c \geq 0$ in case of buy CXL, and $\Delta I \geq 0$ and $\Delta c \leq 0$ in case of sell CXL.

Here, $\tau$ refers to a time scale value, and $p_{mid,\emptyset}$, $\Delta I$, and $\Delta c$ are affected by $\tau$. Therefore, segments are considered to have individual $\tau$ values. By setting a $\tau$ value for each segment, the segment network can be trained.

Each $\pi_{s_i}(o|X;\theta_i)$ can be optimized by standard cross entropy minimization for $\pi(o|X;\theta)$ in Equation (1). The gradient with respect to each $\theta_i$ takes the form

$$\nabla_{\theta_i} CE\left(o_t, \sum_{i=1}^{N} p(s_i|X_t)\pi_{s_i}(o|X_t;\theta_i)\right) \tag{5}$$

where $CE$ is a cross entropy function, and $o_t$ and $X_t$ are the observed order and market states. In addition, the cross entropy gradient with respect to $\theta_s$ can be calculated and added to Equation (3) as an adjustment term. We performed training and validation of the proposed model with and without the adjustment term (we refer to these methods as LSIL1 and LSIL2), and the results may be seen in the experiments section.

## 4. Neural Network Configuration

### 4.1. Feature Engineering

As market states $X$, price series and orderbook features are used.

The price series is comprised of 10 market prices (last or current trade price) taken at certain time step intervals. The time step intervals were selected according to the time scale value of each cluster $\tau_{c_i}$.

The orderbook features are a arranged in a vector array of the latest orderbook volumes at 10 price levels above and below the mid-market price. To distinguish buy and sell orders, buy order volumes are recorded as negative values.

### 4.2. Stock Order Digitization

We consider a single market and three order types: Limit order (LMT), Market order (MKT), and Cancel order (CXL). LMT and CXL specify the market side (i.e., buy or sell), prices, and quantities, and MKT specifies the market side, and quantities.

Orders were digitized based on order type (i.e., LMT, MKT, CXL), order price, and order volume. In practice, some orders are defined as combinations of two or more types of orders. For example, price change orders and volume change orders can be interpreted as a combination of LMT or MKT and CXL. In such a case, the LMT or MKT part of the order that is considered to reflect the latest intentions of the agents, is extracted.

Order price and volume are digitized into possible values. For price, 10 prices above and below the mid price are possible. For volume, up to five times the minimum trading unit is possible, and CXL orders are digitized with negative volume. LMT and CXL orders have $(10 + 10) \times (5 + 5) = 200$ possible values, and MKT orders that do not specify prices have $5 + 5 = 10$ possible values. Thus, 210 values are possible, and orders that do not match any condition are discarded.

### 4.3. Network Architecture

The proposed network architecture is shown in Figure 1. The network consists of the segment network and segment level order networks. Networks outputs are aggregated by Equation (1) and the overall order probability $\pi(o|X)$ is calculated. Reward and order probability losses are calculated using the predicted probabilities, observed order, and reward functions for each segment.

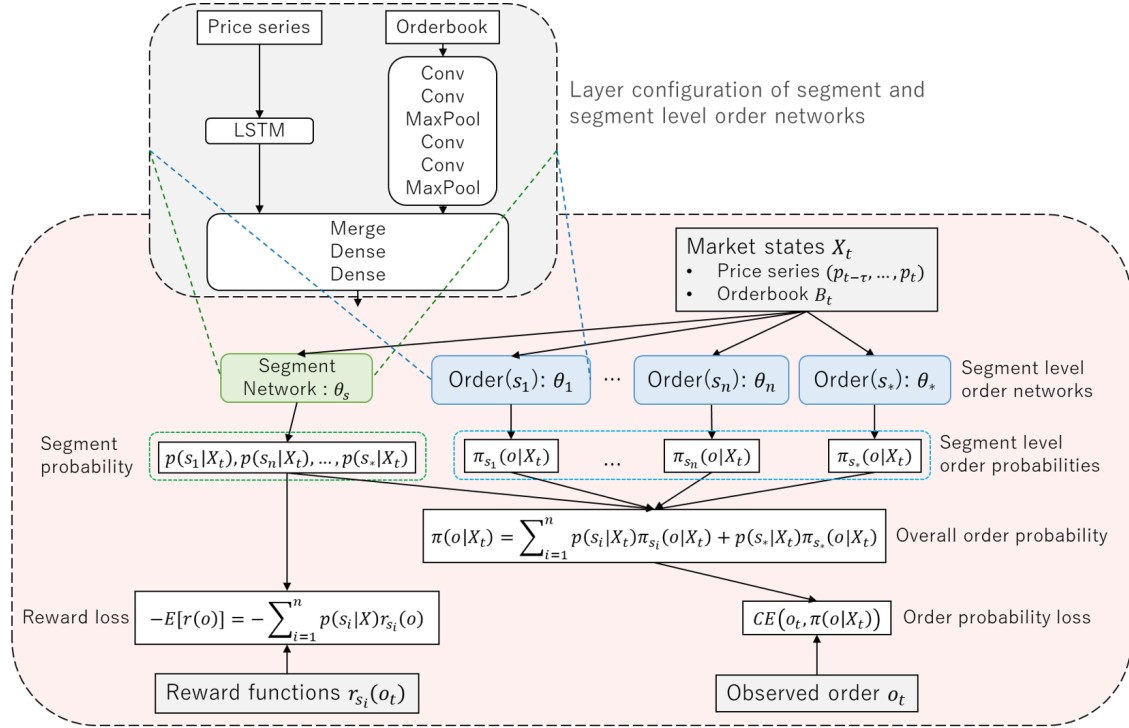

**Figure 1.** Overview of the proposed network. The network consists of the segment network and segment level order networks. The segment network predicts segment probability, and segment level order networks predict segment level order probabilities. The overall order probability is calculated by aggregating segment and segment level order probabilities. Loss functions are calculated with predicted probabilities, observed order, and reward functions for each segment.

Segment network and segment level order networks have the same layer configuration, except for the last fully-connected layer. In these networks, two market state features, price series and orderbook (Section 4.1), are extracted and merged. To extract price series features, a long short-term memory (LSTM) Hochreiter and Schmidhuber (1997) layer is used according to previous studies Bao et al. (2017); Fischer and Krauss (2018). Convolutional layers Krizhevsky et al. (2012) are used to extract orderbook features that have positional information Tashiro et al. (2019); Tsantekidis et al. (2017). Merged features are transformed by fully-connected layers and the segment or order probability is output from the last layer.

## 5. Experiments

Experiments were performed using simulated market data and historical market data. In experiments using simulation data, we ran an artificial market simulation in advance, and trained neural network models using the generated data. The objective of the experiments on artificial data was to verify that the proposed LSIL model could predict segment probability correctly in an idealized setting where order-trader pair information is available. In experiments using historical data, we trained models using actual public stock trading data from the Tokyo Stock Exchange.

The proposed LSIL method was used to train networks with and without the adjustment term (Section 3). We refer to the networks as LSIL1 and LSIL2, respectively, and the proposed method was compared to a standard IL model and GAIL model. The IL model has the same layer configuration as the LSIL segment level order networks and simply predicts order probabilities from market states. The GAIL model is based on sequence generative adversarial nets (SeqGAN) Yu et al. (2017) and generates order sequences without using market states. In addition, a network, which we refer to as segment IL (SIL) with the same network architecture as LSIL, was optimized to minimize only order probability loss and not reward loss.

Model performance was compared using the following benchmarks: precision at k (P@k), area under receiver operating characteristic (AUROC), and expected reward $E[r(o, X)]$. Precision at $k$ is the percentage of correct answers included in the top k classes in predicted scores. P@k and AUROC are calculated for predicted order probabilities. Expected reward $E[r(o, X)]$ is calculated to validate LSIL predicted cluster probability. As reward values are centered, positive $E[r(o, X)]$ indicates that the LSIL model can predict segment probability appropriately.

### 5.1. Experiments on Simulated Data

We ran an artificial market simulation to generate a dataset, and used the dataset to train and validate our LSIL models. The artificial market simulator consists of markets and agents where markets play the role of the environment whose state evolves through the actions of the agents. In each step of simulation, an agent is sampled, the agent submits an order, and markets process orders and update their orderbooks. Market pricing follows a continuous double auction Friedman and Rust (1993).

We define a fundamental price $p_F$ for the market. The fundamental price represents the fair price of the asset/market, is observable by stylized agents and is used to predict future prices. The fundamental price changes according to a geometric Brownian motion (GBM) Eberlein and Keller (1995) process. The volatility of the GBM was set to $5 \times 10^{-6}$.

Stylized agents are commonly used in artificial market simulations to model the behavior of realistic economic actors Hommes (2006), and reproduce many stylized facts of actual financial markets Chiarella and Iori (2002); Chiarella et al. (2009). Stylized agents predict expected future log return $r$ using the following equation:

$$r = \frac{1}{w_F + w_C + w_N}(w_F F + w_C C + w_N N) \tag{6}$$

where

$$F = \frac{1}{\tau} \log(\frac{p_t^*}{p_t}) \tag{7}$$

$$C = \frac{1}{\tau} \log(\frac{p_t}{p_{t-\tau}}) \tag{8}$$

$$N \sim \mathcal{N}(\mu, \sigma^2) \tag{9}$$

and $p_t$ and $p_t^*$ are current market price and fundamental price, respectively, and $\tau$ is the time window size (or time scale). Weight values $w_F$, $w_C$, and $w_N$ are sampled randomly and independently from exponential distributions for each agent. The stylized agents predict future market prices $p_{t+\tau}$ from the predicted log return using the following equation:

$$p_{t+\tau} = p_t \exp(r\tau) \tag{10}$$

A stylized agent submits a buy LMT with price $p_{t+\tau}(1 - k)$ if $p_{t+\tau} > p_t$, and submits a sell LMT with price $p_{t+\tau}(1 + k)$ if $p_{t+\tau} < p_t$. The parameter $k$ is the called order margin and represents the amount of profit that the agent expects from the transaction. In this experiment $k$ was set to 0.01. The submitting volume $v$ is fixed to one.

In this study, the following seven types of stylized agents are registered to the simulator: Type 1 ($15 \leq \tau \leq 25$), Type 2 ($30 \leq \tau \leq 50$), Type 3 ($60 \leq \tau \leq 100$), Type 4 ($120 \leq \tau \leq 200$), Type 5 ($240 \leq \tau \leq 400$), Type 6 ($480 \leq \tau \leq 800$), and Exceptional ($w_f = w_c = 0$). Noise weights $w_N$ were fixed at 0 for for types 1 to 6. To prevent chart term $C$ from becoming too dominant, the expected value of chart weights $w_C$ was attenuated according to $\tau$.

These types of stylized agents reflect our assumption that agents with some type of time scale exist. Here, 100 agents were registered for types 1 to 6 and 400 agents were registered for the exceptional type.

One simulation consists of 101,000 steps where the first 1000 steps were used to build up the initial market orderbook and subsequently discarded. Simulations were performed 10 times with changing random seeds, and the data from the first eight simulations were used for training and the data of the remaining two simulations were used for validation.

According to the configured types of stylized agents, LSIL segments $s_i$ were set as follows: $s_1$: $\tau = 20$, $s_2$: $\tau = 40$, $s_3$: $\tau = 80$, $s_4$: $\tau = 160$, $s_5$: $\tau = 320$, $s_6$: $\tau = 640$, and $s_*$: Exceptional.

The results of modeling are shown in Table 1. For all indicators of order prediction accuracy, the proposed LSIL2 outperformed all other methods. We find that our proposed method worked well without the adjustment term. In addition, since the expected rewards of LSIL1 and LSIL2 were both positive at 0.1127 and 0.0721, we believe LSIL1 and LSIL2 were able to predict segment probabilities appropriately. Appropriate prediction of segment probabilities also contributed to the improvement of prediction accuracy as shown in Table 1.

**Table 1.** Order prediction accuracy for artificial stock order data. Predicted order probabilities are validated with precision at $k = 1, 5, 10$, and AUROC.

| Model | P@1 ↑ | P@5 ↑ | P@10 ↑ | AUROC ↑ |
|-------|-------|-------|--------|---------|
| IL    | 0.0436 | 0.1764 | 0.2889 | 0.8416 |
| GAIL  | 0.0166 | 0.0713 | 0.1391 | 0.8263 |
| SIL   | 0.0546 | 0.2181 | 0.3796 | 0.9071 |
| LSIL1 | 0.0518 | 0.1959 | 0.3446 | 0.9016 |
| LSIL2 | 0.0606 | 0.2439 | 0.4094 | 0.9120 |

## 5.2. Experiments on Historical Data

We used FLEX_FULL historical full-order-book data from the Tokyo Stock Exchange.[1] FLEX_FULL contains tens of millions of stock order data per day recorded in millisecond resolution Brogaard et al. (2014).

In this experiment, data for symbol 9022 (Central Japan Railway Company) collected between 1 January 2018 and 31 December 2018 were used for training, and data collected between 1 January 2019 and 31 August 2019 were used for validation. Training and validation samples were extracted every 10 available samples. The segments $s_i$ of LSIL were set to the same values as the experiments using artificial data.

The average of each segment probability along all validation data is $p(s_1) = 0.4362$, $p(s_2) = 0.0517$, $p(s_3) = 0.0500$, $p(s_4) = 0.0893$, $p(s_5) = 0.1348$, $p(s_6) = 0.1835$, and $p(s_*) = 0.0546$ while the LSIL1 and LSIL2 rewards were 0.0371 and 0.0179. We thus see that traders with the shortest-term rewards are dominant in this market and in agreement with the ratio of orders submitted from the co-location site at the TSE.

The accuracy results are shown in Table 2. We can see that SIL, LSIL1, and LSIL2 predicted orders with similar accuracy. Although LSIL2 outperforms on simulated data, we attribute its underperformance on historical data to the simplicity of our reward function specification. In general, real-market investors are considered to have a wide variety of "reward functions", and therefore more

---

[1]　https://www.jpx.co.jp/english/markets/paid-info-equities/realtime/index.html.

diverse types of reward functions are needed for more accurate prediction. Nevertheless, we are able to obtain salient features of the most dominant segments.

**Table 2.** Order prediction accuracy for historical stock order data. Predicted order probabilities are validated with precision at $k = 1, 5, 10$, and AUROC.

| Model | P@1 ↑ | P@5 ↑ | P@10 ↑ | AUROC ↑ |
|-------|-------|-------|--------|---------|
| IL | 0.1344 | 0.4347 | 0.6266 | 0.9371 |
| GAIL | 0.0456 | 0.2388 | 0.4155 | 0.9091 |
| SIL | 0.1392 | 0.4469 | 0.6381 | 0.9407 |
| LSIL1 | 0.1401 | 0.4466 | 0.6376 | 0.9406 |
| LSIL2 | 0.1398 | 0.4463 | 0.6371 | 0.9404 |

An example of predicted segment probability is shown in Figure 2. It shows changes of segment probabilities predicted using trained LSIL2 model and market prices over time. Segment probabilities fluctuate as market states (market price, orderboook, etc.) change. In this case, we see the market price rose suddenly at the end of the plot, and the probability of $s_1$ temporarily increased just before the rise of the market price. $s_1$ is the segment of traders with the shortest-term rewards. In general, short-term investors are considered to act when price fluctuations are expected immediately afterwards, and the increase in the probability of $s_1$ in Figure 2 is reasonable. Although there are some price fluctuations that are not linked to price fluctuations, we are able to interpret meaningful behavior patterns and gain insight into the agents driving the dynamics of real markets.

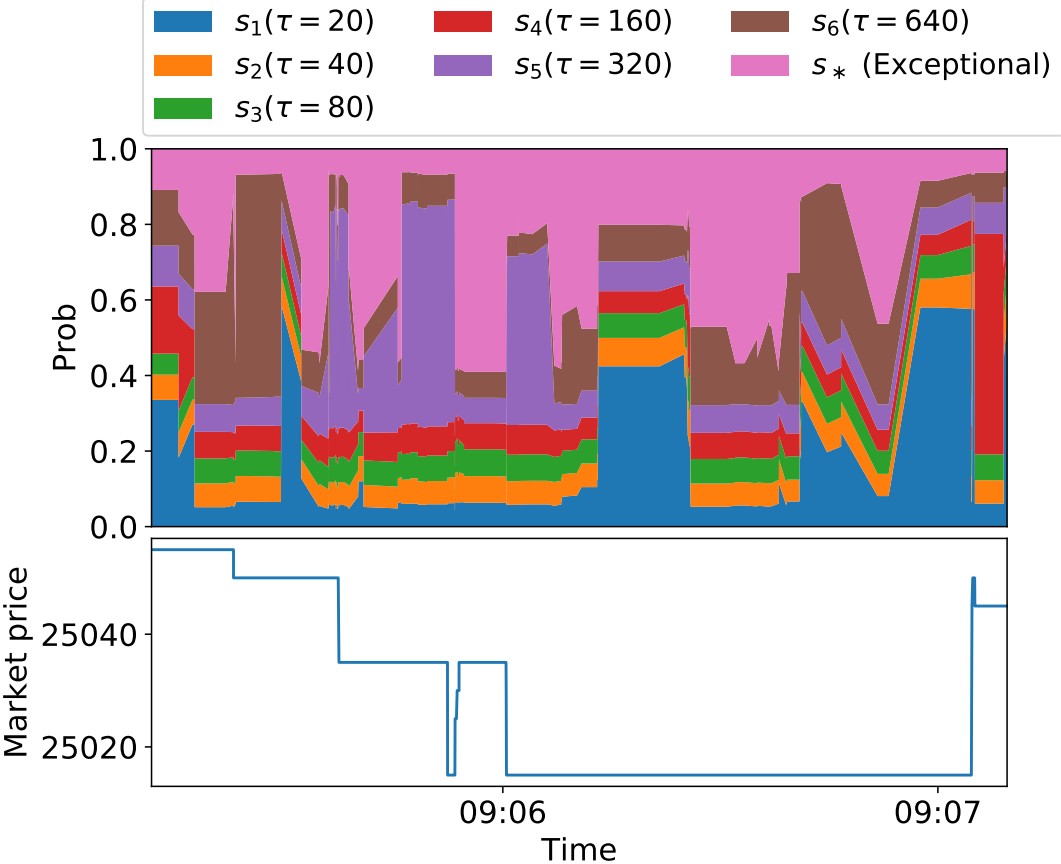

**Figure 2.** Example of predicted segment probabilities using the trained Latent Segmentation Imitation Learning (LSIL)2 model (4 April 2019). The predicted probability of each segment changes over time. In this case, the market price rose suddenly at the end of the plotted period, and the probability of $s_1$ temporarily increased just before the rise of the market price. This probability change indicates a short-term traders action in anticipation of price fluctuations.

## 6. Conclusions

Traders' behavior prediction is an essential issue in financial market research, and it would be very useful if individual investment strategies could be extracted even from anonymized data. The proposed latent segmentation method was able to predict stock order submission probabilities more accurately with latent segmentation compared to naive IL and GAIL. This result shows that the proposed method was able to separate and approximate appropriate strategies. While the predicted segment probabilities by the proposed method are but one of many realizations, we find that our predicted behavior can give useful insight into the profit and loss timescales driving market participants' behaviors under different market scenarios.

The limitation of this study is that the segmentation ability largely depends on the design of the reward function. Real-market investors have a wide variety of objectives Jensen (2001), and rule-based reward functions cannot fully represent them. In future work, we intend to consider a more detailed segmentation based on selection and scaling of reward functions. In addition, we also intend to apply inverse reinforcement learning (IRL) methods Hadfield-Menell et al. (2016); Ng and Russell (2000) for training reward functions from historical trading data. By using IRL, more appropriate reward functions that will improve validity of the segmentation can be obtained.

**Author Contributions:** Conceptualization, I.M.; methodology, I.M., D.d., M.K., K.I., H.S., and A.K.; investigation, I.M.; resources, H.M. and H.S.; writing—original draft preparation, I.M. and D.d.; supervision, K.I.; project administration, K.I. and A.K. All authors have read and agreed to the published version of the manuscript.

**Funding:** This research received no external funding.

**Acknowledgments:** We thank two anonymous reviewers for provided helpful comments on the manuscript.

**Conflicts of Interest:** The authors declare no conflict of interest.

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
