# Peer review of "Latent Segmentation of Stock Trading Strategies Using Multi-Modal Imitation Learning"

_jrfm, doi:10.3390/jrfm13110250_

Round 1

Reviewer 1 Report

The paper covers an interesting and important topic of stock trading strategies modelling. Its overall quality is high as it is based on sound methodology and the results were interpreted correctly. However, there are some issues that should be improved:

  1. Time period of the analysis should be provided in the abstract.
  2. More information about the sample etc. should be presented in Introduction.
  3. The final section ('Conclusions') should definitely be expanded by adding such aspects as: comparison to the results of the previous studies, limitations of the analysis, practical implications and future research directions.
  4. More recent publications should be used.

Author Response

Thank you for your kind comments.  We revised our paper according to your appropriate indication.

  1. Time period of the analysis should be provided in the abstract.
    • We added the time period of the analysis to the abstract.
  2. More information about the sample etc. should be presented in Introduction.
    • We added the sentence "Detailed investigation into changes in market conditions and segments revealed that our proposed segments behaves in line with real-market investor sentiments." to the introduction.
  3. The final section ('Conclusions') should definitely be expanded by adding such aspects as: comparison to the results of the previous studies, limitations of the analysis, practical implications and future research directions.
    • We expanded the final section by adding limitations and future work.  The limitation of this study is that the segmentation ability largely depends on the design of the reward function, and we are going to apply inverse reinforcement learning for training reward functions.
  4. More recent publications should be used.
    • Certainly there was a lack of citations from recent studies.  We added following recent and important studies related to this paper.
      1. Xiao, Shenyong, et al. "Self-evolving trading strategy integrating internet of things and big data." IEEE Internet of Things Journal 5.4 (2017): 2518-2525.
      2. Yu, Lin, Hung-Gay Fung, and Wai Kin Leung. "Momentum or contrarian trading strategy: Which one works better in the Chinese stock market." International Review of Economics & Finance 62 (2019): 87-105.
      3. Chen, Chun-Hao, et al. "An effective approach for obtaining a group trading strategy portfolio using grouping genetic algorithm." IEEE Access 7 (2019): 7313-7325.
      4. Liu, Yang, et al. "Adaptive Quantitative Trading: An Imitative Deep Reinforcement Learning Approach." AAAI. 2020.
      5. Angueira, Jaime, et al. "Exploring the relationship between vehicle type choice and distance traveled: a latent segmentation approach." Transportation letters 11.3 (2019): 146-157.
      6. Villarejo Ramos, Ángel Francisco, Begoña Peral Peral, and Jorge Arenas Gaitán. "Latent segmentation of older adults in the use of social networks and e-banking services." (2019).

Reviewer 2 Report

This article is focused on improving the accuracy of traders' behaviour prediction by using neural network.

Despite relatively low increasing in accuracy, the proposed approach deserves attention from theoretical point of view. In General, i have a good impression after reading this article. The essence is clear, literature review is not exhaustive, but it covers key significant areas, methodology is clearly described.

I have only minor suggestions, mostly related to improving readability and citability of the manuscript:

  1. All figures and tables should be placed after the first mention.
  2. Please, add some results of comparative analysis in the Abstract.
  3. Figure 2, as well as results of comparison require significantly more attention and discussion.
  4. Please, extend the concluding part to show the importance of traders' behaviour prediction and further improving of the proposed method.

Author Response

Thank you for your kind comments.  We revised our paper based on your appropriate indication.

  1. All figures and tables should be placed after the first mention.
    • We fixed the layout of figures and tables.
  2. Please, add some results of comparative analysis in the Abstract.
    • We revised the part of the abstract about experiments as follows; "our results provide interpretable classifications and accurate predictions that outperform other methods in major classification indicators as verified on historical orderbook data from January 2018 to August 2019 obtained from the Tokyo Stock Exchange. By further analyzing the behavior of various trader segments, we confirmed that our proposed segments behaves in line with real-market investor sentiments."
  3. Figure 2, as well as results of comparison require significantly more attention and discussion.
    • We added discussions about Figure 2 and results of comparison.
  4. Please, extend the concluding part to show the importance of traders' behaviour prediction and further improving of the proposed method.
    • We added the contents about the importance of traders' behaviour prediction to the final section.  In, addition, we expanded the final section by adding limitations and future work.  The limitation of this study is that the segmentation ability largely depends on the design of the reward function, and we are going to apply inverse reinforcement learning for training reward functions.